# Evaluation of the Effect of Induced Endotoxemia on ROTEM S^®^ and Platelet Parameters in Beagle Dogs Anaesthetized with Sevoflurane

**DOI:** 10.3390/ani13192997

**Published:** 2023-09-22

**Authors:** Annette P. N. Kutter, Fabiola B. Joerger, Barbara Riond, Barbara Steblaj

**Affiliations:** 1Section of Anaesthesiology, Department for Clinical Diagnostics and Services, Vetsuisse Faculty, University of Zurich, 8057 Zurich, Switzerland; fjoerger@vetclinics.uzh.ch (F.B.J.); bsteblaj@vetclinics.uzh.ch (B.S.); 2Clinical Laboratory, Department for Clinical Diagnostics and Services, Vetsuisse Faculty, University of Zurich, 8057 Zurich, Switzerland; briond@vetclinics.uzh.ch

**Keywords:** canine, d-dimers, hemostasis, platelets, sepsis, thromboaggregometry, thromboelastometry

## Abstract

**Simple Summary:**

Bacterial infection can cause different bleeding or thrombotic disorders in addition to the well-known cardiovascular disorders in dogs. The understanding of the changes of the coagulation system associated with bacterial infection has improved. However, the knowledge of how therapeutic measures against cardiovascular effects might influence the coagulation system is still scarce. The better availability of point-of-care testing in clinics enables the repeated evaluation of coagulation status in clinical patients. This study evaluated the influence of artificially induced endotoxemia and the subsequent treatment with fluids and vasopressors at five timepoints with complete blood count, thrombo-aggregometry and -elastometry and d-dimer analysis in six dogs under sevoflurane anaesthesia. We found signs of hypocoagulability mainly caused by fast decreases in platelet counts. The partial recovery of all coagulation parameters may reflect a positive impact of therapeutic measures of the cardiovascular system on the coagulation status of endotoxemic dogs.

**Abstract:**

Endotoxemia is thought to induce severe changes in coagulation status. In this study, blood samples from six beagle dogs receiving 1 mg/kg *E. coli* lipopolysaccharide (LPS) intravenously were analyzed to describe the concurrent changes in platelet count, platelet function assessed with impedance thromboaggregometry, thromboelastometry and d-dimers during artificially induced endotoxemia and its therapy with fluids and vasopressors at five timepoints (baseline, after LPS and 30 mL/kg Ringer’s acetate, during noradrenaline ± dexmedetomidine infusion, after a second fluid bolus and a second time after vasopressors). Results were analyzed for changes over time with the Friedman test, and statistical significance was set at *p* < 0.05. We found decreased platelet count and function and changes in all platelet-associated rotational thromboelastometry (ROTEM) variables indicating hypocoagulability, as well as increases in d-dimers indicating fibrinolysis within one hour of intravenous administration of LPS, with partial recovery of values after treatment and over time. The fast changes in platelet count, platelet function and ROTEM variables reflect the large impact of endotoxemia on the coagulation system and support repeated evaluation during the progress of endotoxemic diseases. The partial recovery of the variables after initiation of fluid and vasopressor therapy may reflect the positive impact of the currently suggested therapeutic interventions during septic shock in dogs.

## 1. Introduction

Endotoxemia and sepsis are associated with very high mortality rates in people and dogs despite existing guidelines to treat sepsis in people [1]. A prothrombotic state is induced early in sepsis to fight against bacteria [2,3]. This initially prothrombotic state may lead to microthrombosis, organ dysfunction and the depletion of platelets and coagulation factors due to disseminated intravascular coagulation (DIC) with consequent bleeding disorders. In one study, more than 60% of septic dogs showed coagulation disorders, with an increased odds for non-survival when coagulation disorders were diagnosed [4]. In septic dogs, both a decreased and an increased [5,6,7,8] platelet function have been described over the course of the disease, and platelet count is typically decreased [9]. Thromboelastography (TEG) and -metry (TEM) have been used for a more general and faster assessment of coagulation from the initiation of coagulation with a fibrin clot to maximal clot firmness and consequent lysis of the clot [10]. Results of a previous research study in dogs using TEG did show small changes indicative of a more hypocoagulable state 4 hours after low-dose endotoxin [11]. In a larger study in human ICU patients rotational thromboelastometry (ROTEM) variables normalized with improved sequential organ failure assessment (SOFA) scores [12]. In dogs suffering from naturally occurring sepsis, a higher maximal amplitude was a predictor of survival in non-activated TEGs, although none of those TEGs showed signs of hypocoaguability [13]. 

D-dimers have been suggested to be an early indicator of endotoxemia in dogs [11] and were elevated in naturally occurring sepsis [13]. All experimental studies in dogs assessing single parts of the coagulation system after low- and high-dose administration of endotoxin have shown that the hypocoaguable changes in the coagulation system in untreated dogs were sustained for the 24 h duration of the studies [7,9,11]. Experimental studies assessing the changes after 24 h without any treatment were not performed. After high dose endotoxin administration the observed cardiovascular, coagulopathic and kidney damage during the first 6 hours were severe and limit further survival experimental studies [14].

Traditional treatment of coagulation abnormalities includes the administration of pro- and anticoagulatory drugs and blood products. The fast changes in coagulation variables induced by endotoxemia require a repeated and rapid assessment of coagulation abnormalities with point-of-care tests. To the authors’ knowledge, no study has evaluated changes in coagulation in septic patients, in which only cardiovascular and not coagulation changes were treated. 

The aim of this study was to assess changes in hemostasis by describing the devolution in platelet count, platelet function assessed with thromboaggregometry, thromboelastometry and d-dimers during artificially induced endotoxemia and after therapy of hypotension with fluids and vasopressors. 

We hypothesized that coagulation would be impaired after administration of endotoxin and administration of fluids. 

## 2. Materials and Methods

### 2.1. Ethical Standpoint

Dogs of this study were used in three different concurrent and parallel studies. Apart from the current study, the other studies investigated, on the one hand, macro- and microcirculatory variables and fluid responsiveness parameters [15] and, on the other hand, induced kidney [14] injury using two different treatment protocols for endotoxemic shock in a crossover manner described below. The dogs had been scheduled for euthanasia due to the need for postmortem examination as part of a previous study, without impact on our study. Therefore, the potentially lethal studies involving endotoxemia were performed in this group of dogs according to the 3R principles. Ethical approval (ZH057/17) for all studies was obtained from the Cantonal Veterinary Office of Zurich, Switzerland and followed the guidelines of the Swiss Animals Protection Law.

### 2.2. Animals

Six purpose-bred beagle dogs (three intact females and three intact males) with an age of 7.4 (5–9.8) years (median (range)) and a body weight of 13.7 (11.4–17.9) kg were included in this study. On the study days, the dogs were transported to the study site in a travel cage. Previously, they were fasted for 12 h with water available ad libitum. The dogs were found to be healthy based on a preanaesthetic clinical examination, complete blood cell count (CBC), urine, blood gas and serum biochemistry analysis performed at baseline.

### 2.3. Preanaesthetic Preparation, Anaesthesia and Instrumentation

For premedication, 0.2 mg/kg of methadone (Methadon Streuli; Streuli Pharma AG, Uznach, Switzerland) was administered intramuscularly. An intravenous (IV) catheter was aseptically placed in a cephalic vein, anaesthesia was induced with propofol to effect (Propofol 1% Fresenius Kabi AG, Kriens, Switzerland), and orotracheal intubation with a cuffed endotracheal tube was performed. Mechanical ventilation was initiated with a pressure-controlled mode to achieve an end tidal carbon dioxide partial pressure between 35 and 40 mmHg (4.7–5.3 kPa). Anaesthesia was maintained with sevoflurane (Sevorane, AbbVie AG, Cham, Switzerland) to effect in oxygen and air (FIO_2_ 0.6) and a constant rate infusion (CRI) of fentanyl (Fentanyl Sintetica, Sintetica, Mendrisio, Switzerland) at a rate of 0.005 mg/kg/h. All dogs received Ringer’s acetate (RA) (Ringer Acetat, Bichsel AG, Unterseen, Switzerland) at 5 mL/kg/h. Forced-air warming (Bair-Hugger 505, Carbamed, Liebefeld, Switzerland) and a warm-water blanket (T/Pump, Gaymar Industries, New York, NY, USA) were used to maintain normothermia. A urinary catheter, a peripheral arterial catheter and a pulmonary artery catheter to measure cardiac output were placed after aseptic preparation, and cardiovascular measurements were performed as described previously [15]. The catheters were flushed with non-heparinized saline. 

### 2.4. Experimental Design 

After baseline cardiovascular measurements and blood sampling, endotoxemia was induced with 1 mg/kg *E. coli* lipopolysaccharide (LPS) (LPS endotoxin O111:B4; Sigma-Aldrich, Taufkirchen, Deutschland) administered IV over 10 min (Figure 1). Directly after the measurement of macro- and microcirculatory measurements and a CBC, the first phase of fluid resuscitation consisted of 30 mL/kg intravenous administration of RA administered over 30 min. Afterwards, hypotension was treated by titrating IV noradrenaline infusion (Noradrenaline Sintetica, Sintetica) with a 0.0005 mg/kg/minute IV starting dose with or without a 0.0005 mg/kg/h fixed-dose IV dexmedetomidine (Dexdomitor, Provet AG, Lyssach, Switzerland) infusion to reach a mean arterial pressure of 85 to 90 mmHg. Both infusions were stopped thereafter, and when the cardiovascular values had deteriorated after a 10 min washout phase, the fluid resuscitation and vasopressor treatment trial was repeated with or without dexmedetomidine to reach a mean arterial pressure of 85 to 90 mmHg. The dexmedetomidine was given in a randomized blinded manner crossover design to all animals either before 85_1 or before 85_2. All dogs received both treatments. 

The measurements and blood samples were collected at different timepoints: baseline before endotoxin administration (BL), at the end of the first fluid bolus (BL_1), after reaching MAP of 85–90 mmHg in the first trial (85_1), at the end of the second fluid bolus (BL_2) and after reaching MAP of 85–90 mmHg in the crossover trial of the study (85_2) (Figure 1). After LPS, only a CBC was collected. 

### 2.5. Blood Collection and Analysis 

Blood samples were drawn by mild aspiration from the arterial catheter with a 10 mL syringe, and blood collection tubes were filled immediately after sampling in the following order: 3 mL serum tubes (Sarstedt AG, Nürnbrecht, Germany) for serum biochemistry, 2 × 1.3 mL 3.2% sodium citrate tubes (Micro sample tube; Sarstedt) for ROTEM delta and d-dimers, 2 × 1.3 mL lithium-heparin (Micro tube LH; Sarstedt) for ROTEM platelet analysis, 0.5 mL in a blood gas syringe (BD A-line blood gas syringe, Becton Dickinson, Plymouth, UK) for ionized Ca^++^ analysis and 0.5 mL in a K_3_-EDTA tube (Microvette K3E; Sarstedt) for CBC and blood smears. Each tube was inverted carefully several times to allow for a good mixture of blood. 

Blood gas was immediately analyzed on a point-of-care blood gas analyzer (RAPID Point 500, Siemens Healthcare, Zurich, Switzerland), and the EDTA tube was sent to the central laboratory for analysis of CBC (Sysmex-XT 2000iV, Sysmex Cooperation, Kobe, Japan). The analysis of CBC was performed automatically, and each blood sample was controlled for the presence of platelet aggregates by microscopical evaluation by an experienced member of the central laboratory. 

For the ROTEM platelet analysis (TEM innovations GmbH, Munich, Germany), the lithium-heparin blood samples were held at room temperature for at least 15 min. Analysis was performed as described previously [16] with the two different agonists arachidonic acid (ARA-tem) and Adenosindiphosphat (ADP-tem) as reagents for platelet activation. Briefly, diluent (0.9% sodium chloride with preservatives; TEM innovations GmbH) was warmed to 37 °C in the designated heating area prior to analysis. Thereafter, 0.15 mL of diluent and 0.15 mL of whole lithium-heparinized blood were pipetted into a single-use cuvette containing a stirring bar and 2 electrodes (ROTEM platelet cuvette; Tem Innovations GmbH). Each sample was incubated for 3 min while the device determined the impedance baseline value of the system. During this time, 0.02 mL of diluent were added to each of the respective reagents to reconstitute the lyophilized agonist for platelet activation. After determination of a baseline, platelets were activated by addition of 0.012 mL of reconstituted reagent into the cuvette containing whole blood and diluent. Aggregation of the activated platelets leads to increasing electrical impedance, which was recorded over 6 min. The recorded impedance slope was used to calculate the following parameters automatically: area under aggregation curve (AUC; Ω × minute) for overall platelet aggregation, amplitude of measured impedance at 6 min (A6; Ω) for the extent of platelet aggregation and maximum slope (MS; Ω/minute) for the rate of aggregation. 

All ROTEM analyses were performed by one experienced veterinarian (APNK) on a ROTEM delta device (TEM innovations GmbH, Munich, Germany) as previously described [17]. Tests performed included Ex-tem S (extrinsic activation by tissue factor), In-tem S (ellagic acid activation), Fib-tem S (tissue factor activation with cytochalasin D added to block platelets) and Ap-tem S (tissue factor activation with aprotinin added to block fibrinolysis). Sodium citrate tubes were placed on the analyzer’s warming plate and were left for at least 5 min prior to the first analysis and afterwards analyzed as soon as possible. All samples were analyzed for 60 min at 37 °C. The ROTEM tracings were visually evaluated, and parameters were excluded if an artefact was suspected. The following parameters were exported from the ROTEM database and copied into a spreadsheet: clotting time (CT), clot formation time (CFT), alpha angle (α), amplitude at 10 min (A10), maximum clot firmness (MCF), maximum clot elasticity (MCE), maximum lysis (ML) and G, a calculated measure of total clot strength (G = 5000 × MCF/(100 − MCF)). If in any profile MCF did not reach 2 or 20 mm, an infinite CT or CFT was defined as 3600 s. For Fib-TEM S, only CT, MCF, MCE and G were analyzed. A green line in the Fib-TEM tracing was defined as a Fib-TEM-MCF of 1 mm. 

One citrate tube was centrifuged, and the plasma was frozen at −80 °C for batch analysis of d-dimers with a commercially available immunometric flow-through principle assay in the in-house laboratory (D-Dimer single tests, NycoCard Reader II, Abbott AG, Baar, Switzerland)

### 2.6. Statistical Analysis 

Data from the ROTEM database were copied into an Excel sheet. The database was manually checked for errors, and descriptive statistics were performed. All data are presented together with reference intervals [16,18] as median and range (min-max) due to the low number of dogs examined. For the following variables, percent change from baseline was calculated: platelet count, AUC ARA-tem, AUC ADP-tem, MCF Ex-tem, MCF In-tem, MCF Fib-tem, MCF Ap-tem, d-dimers as follows: (Xactual − Xbaseline)/Xbaseline. To assess the platelet contribution to the ROTEM delta results, the difference between MCF Ex-tem and Fib-tem was calculated [19] and is presented as “platelet-tem”.

No power analysis was performed before the study, since the number of dogs was determined by the other study, and the effect of treatment for the concurrent study was unknown beforehand. 

Further statistical analysis of selected parameters was performed with Prism 8.4.2 (GraphPad Software Inc., La Jolla, CA, USA). The data were firstly assessed for normality in their distribution using Shapiro–Wilk tests. Because many of the measured values were not normally distributed, and due to the low number of dogs, non-parametrical tests were chosen for all further statistical tests. To assess the impact of platelet count on the aggregometry results, a Spearman correlation between platelet count and AUC ARA-tem and AUC ADP-tem was performed. Friedman tests were used to analyze changes over time for selected variables only, to reduce the risk of alpha errors. To correct for the impact of the platelet count on aggregometry, the AUC ARA-tem and AUC ADP-tem were divided by the platelet count and presented as AUC/platelet. A *p*-value < 0.05 was set as statistically significant.

## 3. Results

In total, 30 blood samples from six dogs were analyzed with 3/30 ROTEM delta and 6/30 ROTEM platelet tests repeated due to suspected artefacts. The times between blood endotoxin and blood sample collection and the resting times between sample collection and analysis at all timepoints are reported in Table A1. During the trial, ionized calcium measurements (reference interval 1.25–1.4 mmol/L) ranged from 1.14 to 1.38 mmol/L. 

Leucocyte count (reference interval 4700–11,300/µL) decreased (*p* < 0.01) from 5940 (5230–8490)/µL) to median values of <2000/µL during all other timepoints (*p* < 0.01) (Figure 2a). The hematocrit (reference interval in awake animals: 36–54%) increased *(p* < 0.01) from a baseline value of 29 (28–33) % to maximal values of 49 (39–55) % at timepoint 85_1 (Figure 2b). 

### 3.1. Platelet Count and Function

The platelet count (reference interval 150–399,000/µL) decreased from 239,000 (163–312,000)/µL at baseline to 71,000 (53–96,000)/µL during BL1 (*p* < 0.001) and slowly recovered to low normal values during the next timepoints (141,000 (79–172,000/µL) (Figure 3 and Table 1). No platelets aggregates were found in these samples. In the platelet samples collected directly after endotoxin and before the first fluid bolus, little to medium amounts of aggregates were found, and these values were therefore excluded from analysis. Values of this timepoint Endotoxin decreased to 11,000 (8–17,000)/µL. 

Both AUC ARA-tem and AUC ADP-tem changed significantly over time (*p* < 0.01 and <0.05, respectively, Figure 3b,c). There was a significant positive correlation between the platelet count and the thromboaggregometry parameters AUC ARA-tem and ADP-tem (*p* < 0.0001 in Figure 3d and <0.01, respectively,). The calculated values of AUC ARA-tem and ADP-tem divided by the platelet count are also shown (Figure 3e,f). The ROTEM platelet samples were analyzed between 21 (16–47) (ARA-tem) and 22 (19–60) (ADP-tem) minutes after blood sampling (Table A1). 

### 3.2. ROTEM S Parameters 

The ROTEM delta samples were analyzed 24 (5–95) minutes after blood sampling (Table A1), and 26/30 samples were analyzed within 45 min. The four samples analyzed later were all from the two latest timepoints BL2 and 85_2 (dog 2 and 5). All analyzed CTs, CFTs and MCFs except Fib-tem MCF showed significant changes over time (Figure 4 and Table 2). The platelet contribution (Ex-tem MCF–Fib-tem MCF) also changed significantly, whereas changes in maximum lysis in percent did not reach significance. The change in selected thromboelastometry parameters with percent changes is shown in Table 2, and all variables are shown over time in Table A2.

### 3.3. D-Dimer

The d-dimer did increase significantly over time (*p* < 0.01). Median and range are reported in Table 2. 

## 4. Discussion

The current study explored the changes in ROTEM delta coagulation parameters, platelet count, ARA- and ADP-activated platelet function and d-dimers before and after induction of endotoxemia in six healthy beagle dogs. As opposed to studies in clinical patients, no treatment with drugs or blood products was initiated, and only treatment of cardiovascular changes was initiated with a higher arterial blood pressure goal than that proposed by the current human guidelines at the time of the study [1]. All dogs developed thrombocytopenia and changes in different ROTEM variables indicating hypocoagulability as well as increases in d-dimers indicating fibrinolysis after IV administration of 1 mg/kg LPS. Findings are consistent with an activated and partly exhausted coagulation system after induction of endotoxemia. 

### 4.1. Platelet Count and Function 

The intravenous infusion of 1 mg/kg LPS induced a severe decrease in platelet count with partial recovery during fluid and vasopressor administration in all six dogs. A strong and fast decrease in platelet count has been shown in a similar study in 10 awake dogs treated with the same dose of *E. coli* LPS [9]. The platelet count in that study decreased by 73% after 30 min, which is comparable to the median decrease of 69% at BL1 (approximately 1 h after endotoxin) in our study. However, in our study, BL1 was collected after 30 mL/kg fluid therapy with Ringer’s acetate, whereas in the former study, no fluid or other therapy of endotoxemia was given, supporting a platelet decrease during endotoxemia unrelated to blood dilution by fluid bolus therapy. In our study, the values collected directly before fluid therapy around 20 min after the start of endotoxin dropped to very low values with low to medium aggregates, making exact counting impossible. Platelet count remained low for 24 h in the former study, whereas in our study, the decrease in platelet count was stabilized at −41 to −43% after fluid and vasopressor therapy. In another study assessing much lower doses of LPS (0.02 mg/kg), platelet counts also decreased by 85% compared to baseline within 30 min [6]. Platelets play an important role in coagulation, inflammatory and immune processes [2]. Old platelets are removed from the circulation in the liver by the Ashwell–Morrell receptor (AMR), and consequently, thrombopoietin is released to stimulate the production and release of new platelets in the bone marrow [20]. As seen in the current study, this system can adapt to an increased consumption within a very short time. 

It was originally thought that impedance aggregometry is not dependent on platelet count, but in a study in people, the increase in impedance was directly proportional to the number of platelets involved in coating the electrodes by aggregation [21,22]. In the current study, it could also be shown that there was a strong positive correlation of platelet count and both ARA-tem and ADP-tem AUC. A possible explanation is the lack of the platelet-derived agonists ADP and thromboxane when less platelets are available [23]. In another in vitro study in dog blood measuring AUC with Multiplate, the values were decreased when samples were diluted with decreasing leukocyte counts [24]. Furthermore, an increasing hematocrit did also decrease platelet aggregation in people [22]. Possible explanations are ADP metabolism within erythrocytes reducing the ADP concentration in the plasma sample or an artificially increased plasma concentration of heparin in the blood sampling tube with increased red blood cell mass [22]. Hematocrit increased over time in this study, whereas leucocyte count decreased, which may have had an additional negative effect on thromboaggregometry results. 

In two former studies, platelet closure time with PFA-100^TM^ was used as a tool to assess platelet function during endotoxemia with a low (0.02 mg/kg) and high dose (1 mg/kg) of LPS [6,7]. The PFA technique mimics the high shear conditions that are present in vivo, in contrast to the low shear conditions of the aggregometry used in the current study. Within 30 min after LPS administration, an acceleration (shortened closure times at 30 min) was identified despite a concurrent 85% decrease in platelet count, and later (1–8 h after LPS administration), prolonged closure times were found in both tests used (ADP and epinephrine). The authors of the studies assessing platelet closure times proposed that the shortened times could be used as an early indicator of endotoxemia. In our study, no acceleration could be seen; there was only a decrease in function when ARA-tem and ADP-tem AUC were assessed in absolute numbers (Figure 3b,d). However, when the ADP-tem AUC was assessed per platelet (Figure 3f), BL-1 in ADP-tem (after fluid therapy) seemed to show an increased aggregation per platelet similar to the former findings in PFA for ADP and epinephrine, whereas in ARA-tem (Figure 3e), no tendency toward an increased function per platelet could be seen. We developed this calculated parameter for the current study to assess the function of platelets independent of the number. Because it is not an established parameter, and due to the low numbers of dogs, these data are only shown and not statistically analyzed. Our findings suggest that absolute results of impedance aggregometry are probably the result of both platelet number and function. In future studies, the use of this newly developed parameter of AUC/platelet could serve as a possibility to assess platelet function per platelet. 

In a study in critically ill dogs, the negative role of platelet activation and hyperreactivity on survival during the course of the disease could be shown with ADP-activated thromboelastography samples [8]. On the other hand, in a study in dogs with naturally occurring septic peritonitis, a collagen-activated impedance aggregometry AUC below a certain threshold was a predictor of non-survival in dogs, and with ADP and ARA, no thresholds were found [5]. Both hyper- and hyporeactive platelets are a measure of a disbalance of the coagulation system. In the current study, the acute and induced onset of severe endotoxemia led to hypocoagulability in all parameters assessed—at the earliest, one hour after endotoxin (Table A1). The only tendency toward hypercoagulability was found in ADP-tem AUC/platelet. The first measurement after endotoxin was probably too late to catch an early hypercoaguable phase, and with the relatively short duration of the study, we would probably also have missed later hypercoaguable phases.

Broad reference intervals for ARA and ADP-tem AUC are reported and may cause problems in defining what is considered normal [16]. Individual baseline values and evaluation of changes over time have been suggested in order to increase the sensitivity of detecting abnormal values in dogs [25,26]. In the current study, all baseline values were within reference range, and for the next three timepoints, no value after endotoxin was within reference range, so the observed changes in AUC induced by 1 mg/kg endotoxin can be considered relevant, and the reference interval proved to discern between normal and abnormal values. 

### 4.2. ROTEM S Delta 

Overall, the assessed ROTEM S parameters all indicated hypocoagulability induced after the administration of LPS. An early and relevant prolongation of CT was found. This is in contrast to a study assessing the reaction time (R) with Kaolin-activated TEG after low-dose 0.02 mg/kg LPS [11]. In that study, no significant change in R time could be found, and broad overlap with the control group was found. It is not clear whether the differences between studies are attributable to different doses of endotoxin or to the different measurement techniques. In the low-dose study, both prothrombin and activated thromboplastin times were increased after 4 h, which makes an analytical error for the TEG R time in the former study more probable. 

The CFT and alpha angle, which both assess the amplification phase of coagulation, are influenced by both plasmatic and platelet factors. They were changed significantly, especially immediately after endotoxin administration (Table 2 and Table A2). This change was in parallel with the significant decrease in platelet count and function. In the aforementioned study [11], the kinetic time in TEG was slightly increased 1 and 4 h after endotoxin, but large standard deviations and values within the reference interval were reported. We also identified large differences between individuals, but changes over time were significant.

The MCF in all tests containing platelets was decreased, but to a lesser degree than platelet count and function. A median decrease of approximately 25% at the first timepoint in platelet-containing MCFs was identified. ROTEM MCF is considered to be a superior parameter to tests assessing only single parts of the coagulation cascade, as viscoelastic tests such as ROTEM involve many of the factors that are important under in vivo conditions. We therefore consider ROTEM MCF clinically the most relevant of all tests analyzed in the current study. Of note, Ex-tem MCFs of all dogs recovered to normal after treatment with fluids and vasopressors at the end of the study. In a study in people with sepsis, thromboelastometry parameters normalized with decreasing SOFA scores [12]. It seems that improved organ function per se leads to the recovery of coagulation. In the former study, in which dogs received a much lower dose of endotoxin, but in contrast to the current study, no treatment of shock symptoms was administered, both the maximal amplitude and overall clot strength in kaolin-activated TEGs did not return to baseline until 24 h after endotoxin [11]. The recovery of cardiovascular function and decreasing plasma levels of endotoxin over time restored coagulation without any administration of procoagulotory or antifibrinolytic treatments. 

### 4.3. D-Dimers and Maximum Lysis 

D-dimers increased significantly after LPS infusion in the current study. In a study in dogs assessing endotoxemia from 0.5 to 24 h after low-dose endotoxin (0.02 mg/kg) administration, a 2.2-fold increase in d-dimers was reported [11]. Absolute values were highest after 1 h, with another peak at 24 h. In our study including treatment of hypotension, d-dimers were increased after LPS administration. The authors of the previous study judged that d-dimers were the earliest indicator for coagulation abnormalities during endotoxemia. Our findings, with the earliest measurement 1 h after endotoxin and fulminant changes in many coagulation parameters, cannot judge whether d-dimers would have risen earlier than the other variables. The increase in d-dimers should have been associated with an increase in fibrinolytic activity. However, the change in maximum lysis assessed by Ex-tem did not reach significance (Figure 3f). It has been shown previously that septic processes lead to both the enhancement, and later the depletion, of the fibrinolytic system [27]. Identification of hyperfibrinolysis using ROTEM or TEG without tissue plasminogen activator (t-PA) added may underestimate the true incidence of hyperfibrinolysis [28,29] and may be the reason for the discrepancy in our results. With the currently available point of care tests in veterinary medicine, the correct assessment of hyperfibrinolysis is still challenging. 

One dog did not reach the threshold for CT in In-tem, Ex-tem and In-tem with a green line in these tests at timepoint BL1, 85_1 and BL2, with only a small MCF of 8 and 12 mm visible in Ap-tem. In this dog, the absent activation in the lysis-sensitive temograms could be a sign of early and fulminant hyperfibrinolysis inhibiting the formation of even the smallest clot in vitro. 

### 4.4. Limitations

The current study faced some limitations: The sample size in this study was determined by another study and was kept low to reduce the number of animals being euthanized (3R principles). It is possible that the current study was underpowered to show small effects and to describe changes over time in detail; however, the changes in platelet count, ROTEM platelet, ROTEM delta and d-dimers were large and uniform enough between individuals to reach statistical significance in many parameters. However, due to the low number of animals and the high number of timepoints no post-hoc pairwise comparisons are described in the current study. 

Additionally, we were not able to measure either ROTEM platelet or delta in duplicate, because only one device was available. We relied on normal test performance and only repeated the tests if curves showed no activation at baseline or when the curve had a much flatter slope than the one with the other activator at the same timepoint. It is therefore possible that single measurement errors went undetected. However, the changes in results of the ARA-tem and ADP-tem were similar among individuals over all timepoints, and all evaluated samples at baseline were within reference ranges [16]. For ROTEM delta, all curves were controlled for artifacts and unusual appearance and repeated if deemed necessary. A recent publication that was published after this study was performed has shown that repeatability of CT and CFT values in dogs was low, and the repeatability of MCF (especially In-tem) was good [30]. In the current study, all CTs, CFTs and MCFs similarly changed over time, and therefore, we judge the results as relevant. 

The resting time of the ROTEM delta samples had a tendency to increase (Table A1) during the study due to the one-hour duration of measurements in the ROTEM delta device compared to shorter times between blood collections. In a study comparing reference intervals for different resting times, a tendency to become hypocoaguable was found in ROTEM delta samples resting 40 or 70 compared to 10 min [31]. We cannot exclude that some of the observed changes in ROTEM delta were caused by increasing the resting time of the samples at later time points (BL2 and 85_2, Table A1). However, as the most severe decrease in MCF happened at BL1 (and resting times were not longer than at baseline), the influence of resting time was probably of minor importance. For the ROTEM platelet sample, resting time did not increase over time due to shorter measurement times per sample (6 min compared to 60 min with ROTEM delta) and was always lower than 60 min. For the observed changes in ROTEM platelet variables, the different resting times were therefore not deemed important. In a pilot study in seven healthy dogs, lithium-heparin sample ARA-tem and ADP-tem AUC values had a tendency to increase slightly between 10 and 60 min resting time [16], which is in contrast to ROTEM delta variables that became more hypocoaguable over time in a recent study [31].

Due to the goal-directed definition of timepoints (blood pressure), no uniform times between blood samples were achieved. The time ranges are reported in results (Table A1). The first time point, which proved to show the most severe changes, was 57 (49–65) minutes after the start of endotoxin infusion, and results can therefore be compared to former studies. Additionally, blood sampling was performed under anaesthesia, which can explain the decrease in hematocrit found at baseline. In a former study in beagle dogs, no relevant influence of sevoflurane anaesthesia on ROTEM S variables could be found, and hematocrit was also decreased [32]. Additionally, all blood samples were taken under the same dose of anaesthetic in the current study, which would have no influence on the assessed changes over time. The goal of the concurrent study was to test the effect of the coadministration of dexmedetomidine on macro- and microcirculation. It has been shown that dexmedetomidine has an anti-inflammatory effect in dogs [33]. In the current study, we were not able to test the influence on coagulation changes due to an insufficient sample size, but we cannot exclude the possibility that we missed such an effect. 

As a further important limitation, the current results were found in dogs in an endotoxaemia model and not in naturally occurring sepsis. This limits the value of the results for the clinical patient. On the other hand, the experimental trial allowed to examine the influence of endotoxaemia on the coagulation system under well-defined conditions. 

## 5. Conclusions

In the current study, all dogs developed thrombocytopenia and changes in different ROTEM variables indicating hypocoagulability, as well as increases in d-dimers indicating fibrinolysis within one hour of intravenous administration of LPS. The fast changes in platelet count, platelet function and thromboelastometry variables reflect the large impact of endotoxemia on the coagulation system and support repeated evaluation during the progress of endotoxemic diseases. The partial recovery of the variables after the initiation of fluid and vasopressor therapy may reflect the positive impact of the currently suggested therapeutic interventions during septic shock in dogs. 

## Figures and Tables

**Figure 1 animals-13-02997-f001:**
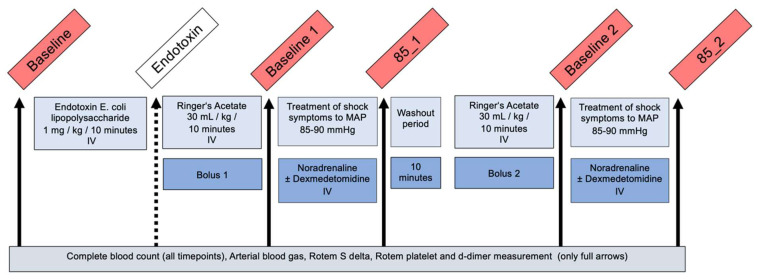
Study procedure: The six blood samples were collected in 6 beagle dogs under sevoflurane anaesthesia at the following timepoints: baseline (BL) after instrumentation, after 1 mg/kg intravenous (IV) *E. coli* lipopolysaccharide infusion (only complete blood count), baseline 1 (BL_1) after endotoxin and 30 mL/kg Ringer’s acetate IV, 85_1 after reaching mean arterial blood pressure 85–90 mmHg with noradrenaline ± dexmedetomidine IV infusion, after another 30 mL/kg Ringer’s acetate IV = baseline 2 (BL_2) and 85_2 after reaching mean arterial blood pressure 85–90 mmHg with noradrenaline ± dexmedetomidine IV infusion in a randomized blinded crossover order.

**Figure 2 animals-13-02997-f002:**
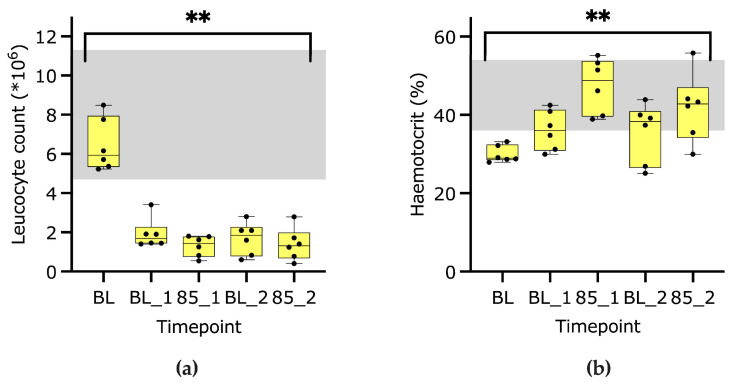
Box-plot graphs of leucocyte count (**a**) and hematocrit (**b**) analyzed at baseline (BL), baseline 1 (BL_1), mean arterial blood pressure 85–90 mmHg (85_1), baseline 2 (BL_2) and 85_2. ** *p*-value < 0.01, and the grey area marks the reference interval.

**Figure 3 animals-13-02997-f003:**
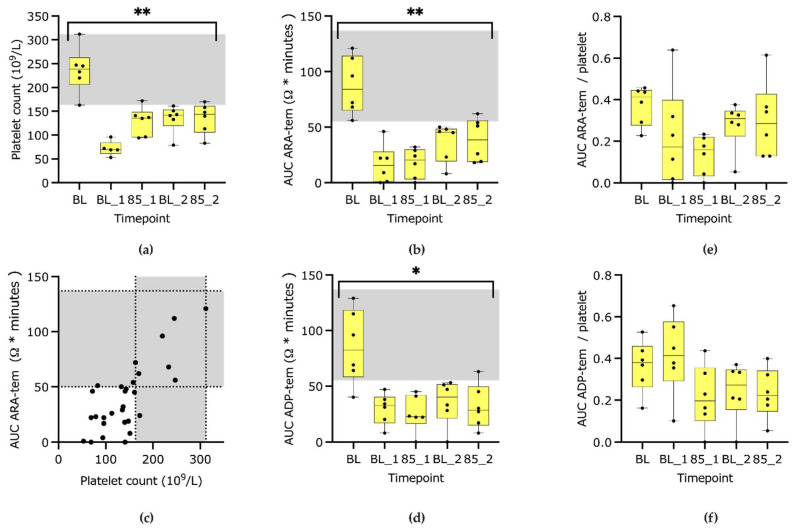
Box-plot graphs of platelet count (**a**), area under the curve (AUC) of arachidonic acid-activated (ARA)-tem (**b**) and their correlation (**c**), AUC Adenosindiphosphat-activated (ADP)-tem (**d**), AUC ARA-tem/platelet (**e**) and AUC ADP-tem/platelet (**f**) analyzed at baseline (BL), baseline 1 (BL_1), mean arterial blood pressure 85–90 mmHg (85_1), baseline 2 (BL_2) and 85_2. * marks a *p*-value < 0.05, ** <0.01, and the grey area marks the reference interval. The data for AUC/platelet were not analyzed statistically.

**Figure 4 animals-13-02997-f004:**
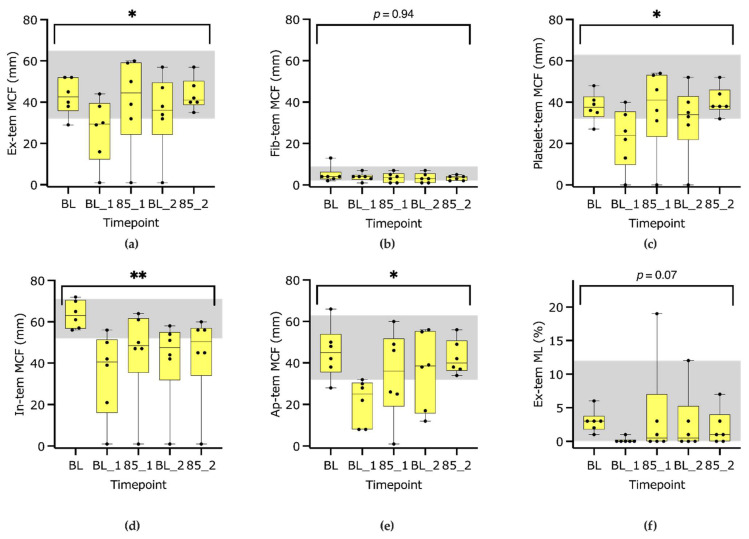
Box-plot graphs of Ex-tem maximum clot firmness (MCF) (**a**), Fib-tem MCF (**b**), Platelet -tem (Ex-tem–Fib-tem) MCF (**c**), In-tem MCF (**d**), Ap-tem MCF (**e**) and Ex-tem maximum lysis (ML) (**f**) analyzed at baseline (BL), baseline 1 (BL_1), mean arterial blood pressure 85–90 mmHg (85_1), baseline 2 (BL_2) and 85_2. * marks a *p*-value < 0.05, ** <0.01, and the grey area marks the reference intervals.

**Table 1 animals-13-02997-t001:** Platelet count and platelet function parameters over time.

Test	Parameter	Timepoint	Reference Interval	Median	Min	Max	Friedmann Test *p*-Value	Median % Change to Baseline
Complete blood count	Platelet count (10^9^/L)	Baseline	150–399	239	163	312	**<0.001**	0%
		BL1		71	53	96		−69%
		85_1		136	94	172		−43%
		BL2		142	79	161		−41%
		85_2		144	83	170		−43%
ARA-tem	AUC (Ω × minutes)	Baseline	55–137	84	56	121	**<0.01**	0%
		BL1		16	0	46		−82%
		85_1		21	0	32		−74%
		BL2		46	8	50		−61%
		85_2		39	18	62		−59%
	A6 (Ω)	Baseline	15–35	23	19	31	np	0%
		BL1		5	0	14		−74%
		85_1		4	0	8		−80%
		BL2		12	1	14		−55%
		85_2		11	2	17		−61%
	Maximal Slope (Ω/minute)	Baseline	4–12	7	5	12	np	0%
		BL1		2	0	3		−83%
		85_1		3	0	3		−65%
		BL2		3	1	5		−65%
		85_2		3	2	4		−60%
ADP-tem	AUC (Ω × minutes)	Baseline	49–137	83	40	129	**<0.05**	0%
		BL1		33	8	47		−69%
		85_1		23	0	45		−65%
		BL2		40	0	53		−67%
		85_2		29	8	63		−73%
	A6 (Ω)	Baseline	12–34	24	12	31	np	0%
		BL1		10	3	13		−63%
		85_1		5	0	11		−68%
		BL2		11	0	13		−65%
		85_2		7	3	16		−73%
	Maximal Slope (Ω/minute)	Baseline	4–13	6	3	13	np	0%
		BL1		2	1	4		−75%
		85_1		2	0	4		−55%
		BL2		3	0	5		−67%
		85_2		2	1	5		−69%

ARA-tem—arachidonic acid-activated test; ADP-tem—Adenosindiphosphat-activated test; AUC—area under the curve; A6—amplitude of measured impedance at 6 min; BL1—baseline 1; 85_1—mean arterial blood pressure 85–90 mmHg; BL_2—baseline 2 and 85_2—mean arterial blood pressure 85–90 mmHg. Significant *p*-values are presented in bold. np—not performed.

**Table 2 animals-13-02997-t002:** Selected In-tem S, Ex-tem S, Fib-tem S, Ap-tem S parameters and d-dimers over time.

Test	Parameter	Timepoint	Reference Interval	Median	Min	Max	Overall *p*-Value Friedmann Test	Median % Change to Baseline
In-tem	CT (sec)	Baseline	133–210	178	150	228	**<0.05**	0%
		BL1		256	190	3600		26%
		85_1		252	187	3600		35%
		BL2		239	97	3600		33%
		85_2		223	190	3600		22%
	CFT (sec)	Baseline	59–201	58	40	99	**<0.01**	0%
		BL1		221	85	3600		118%
		85_1		169	63	3600		90%
		BL2		159	78	3600		135%
		85_2		134	68	3600		99%
	MCF (mm)	Baseline	52–71	63	56	72	**<0.001**	0%
		BL1		41	1	56		−26%
		85_1		49	1	64		−17%
		BL2		48	1	58		−24%
		85_2		51	1	60		−20%
Ex-tem	CT (sec)	Baseline	23–87	35	26	69	**<0.05**	0%
		BL1		53	23	3600		−12%
		85_1		53	31	3600		38%
		BL2		81	31	3600		73%
		85_2		65	33	77		47%
	CFT (sec)	Baseline	85–357	196	129	519	**<0.05**	0%
		BL1		579	184	3600		123%
		85_1		175	78	3600		−15%
		BL2		308	103	3600		21%
		85_2		189	117	270		−17%
	MCF (mm)	Baseline	32–65	43	29	52	**<0.05**	0%
		BL1		30	1	44		−24%
		85_1		45	1	60		6%
		BL2		36	1	57		−13%
		85_2		41	35	57		5%
Fib-tem	MCF (mm)	Baseline	2–9	4	2	13	=0.94	0%
		BL1		4	1	7		0%
		85_1		4	1	7		−23%
		BL2		3	1	7		−31%
		85_2		4	2	5		−13%
Ap-tem	MCF (mm)	Baseline	32–63	45	28	66	**<0.05**	0%
		BL1		25	8	32		−54%
		85_1		36	1	60		−10%
		BL2		39	12	56		−12%
		85_2		40	34	56		−6%
D-dimers	mg/L	Baseline	0–0.5	0.01	0.01	0.10	**<0.01**	0%
		BL1		0.15	0.01	0.60		900%
		85_1		0.40	0.10	1.40		2100%
		BL2		0.20	0.01	0.80		1300%
		85_2		0.30	0.20	1.70		1900%

Min—lowest measured value; Max—highest measured value; In-tem—ellagic acid-activated temogram; Ex-tem—tissue factor-activated temogram; Fib-tem—tissue factor-activated temogram with platelet inhibition; Ap-tem—tissue factor-activated temogram with lysis inhibition; CT—clotting time; CFT—clot formation time; BL1—baseline 1; 85_1—mean arterial blood pressure 85–90 mmHg; BL_2—baseline 2 and 85_2—mean arterial blood pressure 85–90 mmHg. Significant *p*-values are presented in bold.

## Data Availability

Data are available on request.

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
