# Peer review of "Evaluation of the Effect of Induced Endotoxemia on ROTEM S® and Platelet Parameters in Beagle Dogs Anaesthetized with Sevoflurane"

_animals, 2023, doi:10.3390/ani13192997_

Round 1

Reviewer 1 Report

Please, read the attached file.

Minor editing of English language required.

Author Response

Dear Reviewer 1

please see our uploaded review response. thank you.

Reviewer 2 Report

There is a lack of knowledge about coagulopathy in canine and feline medicine. The manuscript is very interesting although the small number of animals and not  inconclusive results, contributes a lot to the understanding of sepsis processes.

In line 31 should be information how dogs receive LPS. IM?

Line 35 , after LPS? How long after? 30 min after 1 hour?

In line 39 ROTEM is used the first time-should be explained.

In fig 1 ad “ 1mg/kg iv”

In line 490 the low haematocrit could be cause be the fluidotheraphy. Should be explained the difference between result after fluidotheraphy and LPS.

Author Response

In line 31 should be information how dogs receive LPS. IM?

- the term intravenously was added.

Line 35 , after LPS? How long after? 30 min after 1 hour?

it is a bit too complicated for the abstract time wise and will not fit into the 200 Word count. The  times and  ranges are recorded in the appendix table 1 to give an idea of timing. However I would be happy to add more detailed information but I am not sure which information to omit from the abstract to fit into the word count.

In line 39 ROTEM is used the first time-should be explained.

Thanks for pointing this out. It was defined in abstract at first time as well as in text (line 64).

In fig 1 ad “ 1mg/kg iv”

Done, I changed the picture and the legend and added IV everywhere where drugs were given IV.

In line 490 the low haematocrit could be cause be the fluidotheraphy. Should be explained the difference between result after fluidotheraphy and LPS.

We added a figure with the haemotocrit results, which shows that the haemotocrit was raising and not decreasing after fluidotherapy. This was with the old version of manuscript not clearly pointed out. As the reviewer 1 also was raising questions about these results we decided to go more into detail there. please also see the answers to this reviewer one.